# Epidemiology and biology of a herpesvirus in rabies endemic vampire bat populations

Megan E. Griffiths [1,2 ✉], Laura M. Bergner [1,2], Alice Broos [1,2], Diana K. Meza[1,2], Ana da Silva Filipe [1], Andrew Davison [1], Carlos Tello[3,4], Daniel J. Becker[5] & Daniel G. Streicker [1,2]

Rabies is a viral zoonosis transmitted by vampire bats across Latin America. Substantial public health and agricultural burdens remain, despite decades of bats culls and livestock vaccinations. Virally vectored vaccines that spread autonomously through bat populations are a theoretically appealing solution to managing rabies in its reservoir host. We investigate the biological and epidemiological suitability of a vampire bat betaherpesvirus (DrBHV) to act as a vaccine vector. In 25 sites across Peru with serological and/or molecular evidence of rabies circulation, DrBHV infects 80–100% of bats, suggesting potential for high population-level vaccine coverage. Phylogenetic analysis reveals host specificity within neotropical bats, limiting risks to non-target species. Finally, deep sequencing illustrates DrBHV super-infections in individual bats, implying that DrBHV-vectored vaccines might invade despite the highly prevalent wild-type virus. These results indicate DrBHV as a promising candidate vector for a transmissible rabies vaccine, and provide a framework to discover and evaluate candidate viral vectors for vaccines against bat-borne zoonoses.

[1] MRC–University of Glasgow Centre for Virus Research, Glasgow, UK. [2] Institute of Biodiversity, Animal Health and Comparative Medicine, College of Medical, Veterinary and Life Sciences, University of Glasgow, Glasgow, UK. [3] Association for the Conservation and Development of Natural Resources, Lima, Perú. [4] Yunkawasi, Lima, Perú. [5] Department of Biology, Indiana University, Bloomington, IN, USA. ✉email: m.griffiths.1@research.gla.ac.uk

Common vampire bat (*Desmodus rotundus*)-transmitted rabies virus (VBRV, species *Rabies lyssavirus*, genus *Lyssavirus*, family *Rhabdoviridae*) is almost invariably fatal in mammals, including humans[1]. Across Latin America, VBRV mortality in livestock is estimated to cause losses of over US$30 million per year[2], and in many countries the number of vampire bat-to-human rabies cases now surpasses those attributed to dogs or wild carnivores[3]. Strategies to limit the burden of VBRV have had limited success[4]. Human vaccination coverage is poor in the geographically isolated communities where outbreaks are most pervasive, and, although pre-exposure vaccination of livestock is effective, it is not sufficiently widely adopted to prevent major losses owing to the relatively high costs to subsistence farmers[5]. In most Latin American countries, vampire bat populations are also controlled using anticoagulant poisons[6]. Culls reduce populations of vampire bats but mathematical models and field experiments suggest that appreciable reduction of the burden of VBRV might require coordinating intensive culls across impractically large geographical areas[7–10].

Vaccination of wildlife reservoirs is a tried and tested method of rabies control in terrestrial reservoir hosts. Large-scale rabies vaccination programmes using oral vaccines distributed in baits successfully controlled or eradicated rabies in much of Europe and North America[11,12]. Despite the resounding success of these vaccination campaigns, similar attempts to target bats, the primary source of human rabies exposure in much of the Americas, face unresolved challenges. Primary amongst these is finding a method of scalable vaccine delivery that achieves sufficiently high population-level coverage in remote and often inaccessible wild bat populations to alter viral transmission dynamics.

Vaccines that spread unaided between individuals ('transmissible vaccines') have been proposed as a potential solution, since high population level coverage might be attained from a limited number of initial deployment efforts[13]. Such vaccines are increasingly feasible to generate in the laboratory by engineering naturally present, replication competent viruses as vectors to express immunogenic proteins from the pathogenic virus of interest (hereafter, 'target virus')[14]. Given high degrees of social interaction that would promote vaccine spread and long lifespans and low reproductive rates that would preserve population-level immunity, transmissible vaccines could be particularly effective in many bat species, including vampire bats.

To date, real-world applications of transmissible vaccines have been hampered by safety and efficacy constraints rooted in virus biology, with only one reaching field trials[13]. In contrast to non-transmissible vaccines where virulence can be genetically attenuated, transmissible vaccine vectors may need to be inherently non-pathogenic, since sustained spread would pose unacceptably high risks of reversion to virulence (e.g., as observed during unintended spread of live-attenuated polio vaccines[15]). To ensure safety in non-target species, transmissible vaccines should be host specific[16], and to be effective, they must reach high population-level prevalence of infection. More specifically, a key theoretical criterion for eradication of the target virus is that the basic reproductive number of the vaccine must exceed that of the target virus[17]. High vaccine transmissibility may be difficult to achieve for at least two reasons. First, the genomic engineering that converts wild-type viruses into vaccines is likely to reduce transmission[14]. Second, immunological cross-reactivity of viral vectors with already circulating wild-type viruses may impede vaccine invasion[18]. Ideal vectors should therefore be benign, commonly occurring, host-specific viruses that can infect individuals that have already been infected by the wild-type version of the viral vector (i.e., capacity for 'super-infection')[14,15,19].

One group of viruses that offers promising transmissible vaccine vector candidates is the subfamily *Betaherpesvirinae* in the family *Herpesviridae* (known as betaherpesviruses, BHVs). These large, double-stranded DNA viruses are typically ubiquitous in their hosts, causing lifelong latent infections with occasional reactivation, e.g. human cytomegalovirus (HCMV)[20]. BHVs are largely host-specific, and healthy adults usually show no overt signs of disease. BHVs can also reach high population-level prevalence are transmitted via multiple routes in bodily fluids (faecal matter, saliva, etc.), and possess a remarkable capability for super-infection[18,20–22]. Finally, BHV vectors induce potent immunological responses that are often equally or more powerful and longer lasting than those induced by natural infections with target viruses[23]. The immune response is also potentially self-boosting given the propensity for latency and reactivation of herpesviruses. As a result, BHV vectors have been successfully developed to target a variety of pathogens. These include murine cytomegalovirus-vectored vaccines for *Mycobacterium tuberculosis*[24] and rhesus macaque cytomegalovirus-vectored vaccines for *Plasmodium* malaria[25], Ebola, and HIV/SIV[26–28]. However, none of these examples currently utilise the transmissible potential of BHVs.

Bats maintain BHVs globally[29–31], opening the tantalising possibility that these viruses might be exploited to vaccinate bat reservoirs against zoonotic viruses, pre-empting their emergence in humans or domestic animals. In the context of VBRV, virally vectored vaccines have been widely discussed as an alternative to culling. However, existing vectors, including vaccinia virus and raccoonpox virus, infect diverse mammalian species, including humans[32–35] and therefore have deliberately attenuated capacity for shedding and transmission. Hypothetical BHV vectors may offer a solution: a host-specific, transmissible rabies vaccine. Although BHVs in *D. rotundus* have not been formally described, a recent metagenomic study reported sequences related to BHVs in several pools of saliva samples from Peru[29].

Here, we report field and genomic studies describing this virus (hereafter DrBHV) and assess its suitability for further study as a vector for a transmissible rabies vaccine. Specifically, we address three central objectives: (1) to quantify BHV prevalence across populations and demographic groups of vampire bats in Peru, (2) to examine VBRV seroprevalence, prevalence and distribution to determine the potential use of a BHV vector for targeting rabies, (3) to use phylogenetic analyses of DrBHV and other BHVs to investigate the host specificity of DrBHV, and (4) to use deep sequencing to evaluate the potential for DrBHV super-infections in individual bats and obtain a whole genome sequence to evaluate the structural similarity of DrBHV to already developed BHV vectors.

## Results

**High prevalence of BHV infection in *D. rotundus*.** We used a semi-nested PCR of the highly conserved BHV UL89 to assess rates of BHV infection in 21 bat species ($N = 111$ individuals) that co-roost with *D. rotundus*, as well as 128 *D. rotundus* individuals from 22 sites across Peru. This sampling revealed BHV infection in nine bat species, spanning all three families of New World bats sampled: Phyllostomidae, Vespertilionidae, and Emballonuridae (Fig. 1).

Relative to the other New World bats that we sampled and results from previous studies in other bat species, BHVs were detected at particularly high frequency in *D. rotundus*, even after controlling for sample size differences among species (Fig. 1, Supplementary Table 1 and 2). In total, 96.9% (124/128) of the vampire bat saliva samples were BHV-positive by semi-nested PCR, with 46.9% (60/128) positive after a single round of PCR. All age classes, adults ($N = 74$), juveniles ($N = 39$) and sub-adults ($N = 15$), were infected. A Generalised Linear Mixed Model

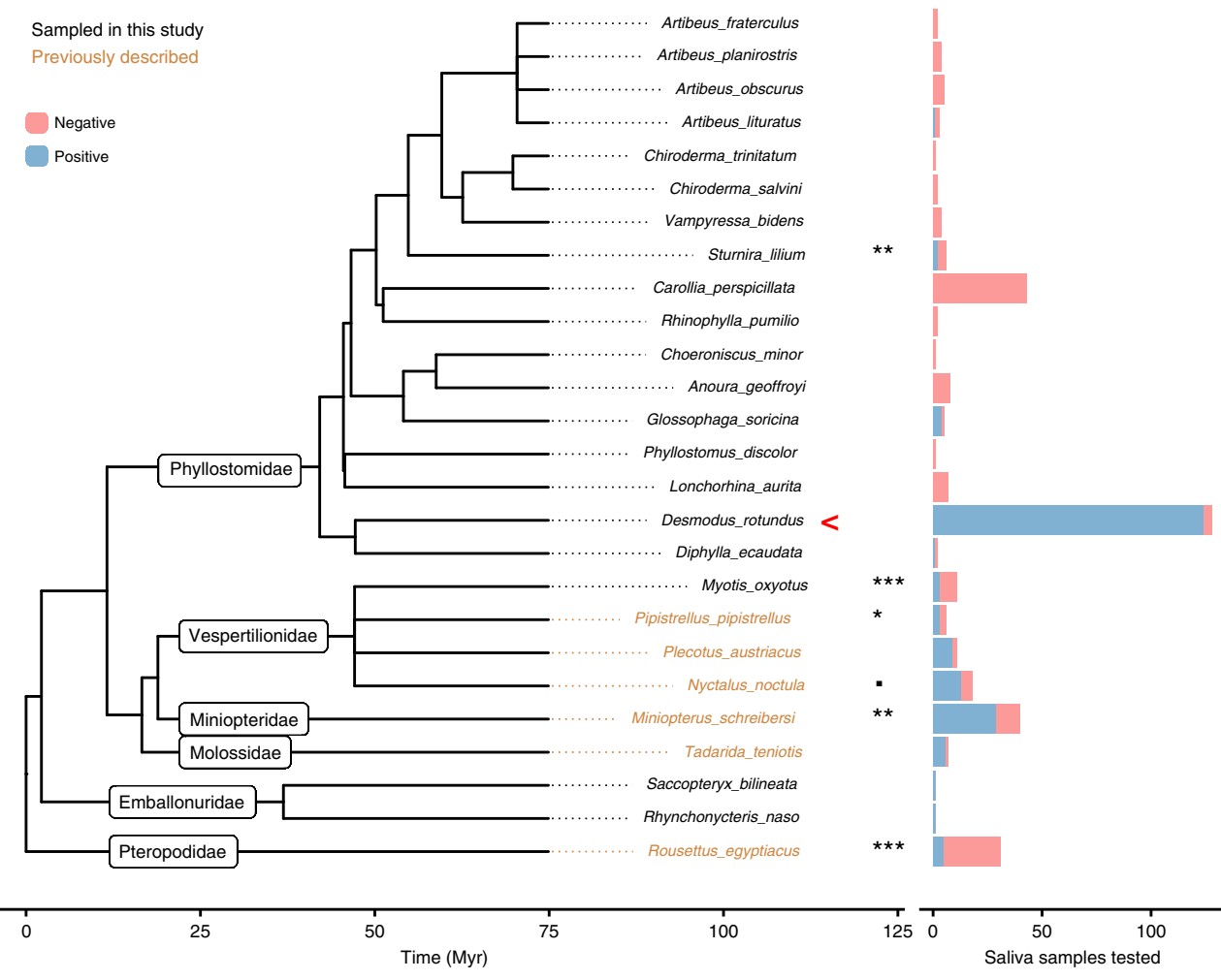

**Fig. 1 Widespread BHV infection across the bat phylogeny.** The bat phylogeny (left) was extracted from the mammalian super-tree to contain species that were tested here or elsewhere for BHV infection[30,71]. Labels on tree branches indicate bat families. Taxa in black text indicate bat species tested here for the first time. Stacked bar charts (right) indicate the number of saliva samples that tested either positive or negative for BHV by PCR of the terminase gene UL89. A binomial generalised linear mixed model (GLMM) followed by post-hoc pairwise comparisons of estimated prevalence by species (multiple comparisons of means: Tukey contrasts, two tailed) was used to test the significance of differences in prevalence. The pairwise significance of prevalence in each species compared to *D. rotundus* is shown on the bar chart with corrected *P*-values: > 0.1, * > 0.05, ** > 0.01, *** > 0.005. All other pairwise comparisons were non-significant. *D. rotundus* is indicated with a red arrow. Source data are provided as a Source Data file.

(GLMM) showed that none of the fixed or random effects tested (site, age, sex, eco-region, year, season and colony structure—vampire bats only, or co-roosting with other bat species) significantly affected BHV detection (Supplementary Table 3). Three bats were sampled at multiple time points from 2015–2018, and 3/3 were positive at all time points. BHVs were also detected in 40% (17/41) of *D. rotundus* blood samples. All 11 bats for which paired samples (both saliva and blood) were tested were positive in saliva, and 45% (5/11) were also positive in blood, confirming systemic infection of bats rather than potential (but unlikely) contamination of saliva from feeding. A lower prevalence in blood compared to saliva has also been observed in HCMV, as viremia is sporadic in actively infected individuals[36].

**Widespread distribution of VBRV in *D. rotundus*.** Eradicating zoonoses using transmissible vaccines requires that vaccines transmit better than the targeted zoonosis[17]. We therefore evaluated the prevalence of active VBRV infection (viral shedding) and past exposure (virus neutralising antibodies) in wild vampire bats across Peru. Of the 128 individuals from which saliva samples were tested for DrBHV, 123 could also be tested for the presence of VBRV RNA using a nested reverse transcription (RT) PCR targeting the nucleoprotein gene. The prevalence of active rabies infection among vampire bats was low (0.8%), with only one individual RT-PCR positive (Bat ID: 6024, Site: CAJ4, Year: 2016). Sequencing this amplicon and BLASTn showed that this virus was most genetically similar to viruses we previously sequenced from livestock in this department of Peru[37]. Among 99 of the individuals previously tested by BHV PCR, with serum samples available, 12 (12.1%) had rabies virus neutralising antibodies > = 0.166 International Units (Supplementary Data 1). Analysis of a larger set of vampire bat sera from 2015 to 2017[38] revealed spatial and temporal variation in seroprevalence (ranging from 0 to 35%) in areas of Peru where rabies outbreaks occur in livestock, presumably reflecting the spatial metapopulation dynamics that underpin VBRV endemicity[7,39] (Fig. 2).

**Phylogenetic separation and host specificity of bat BHVs.** Bayesian phylogenetic analysis of the UL89 nucleotide (304 bp)

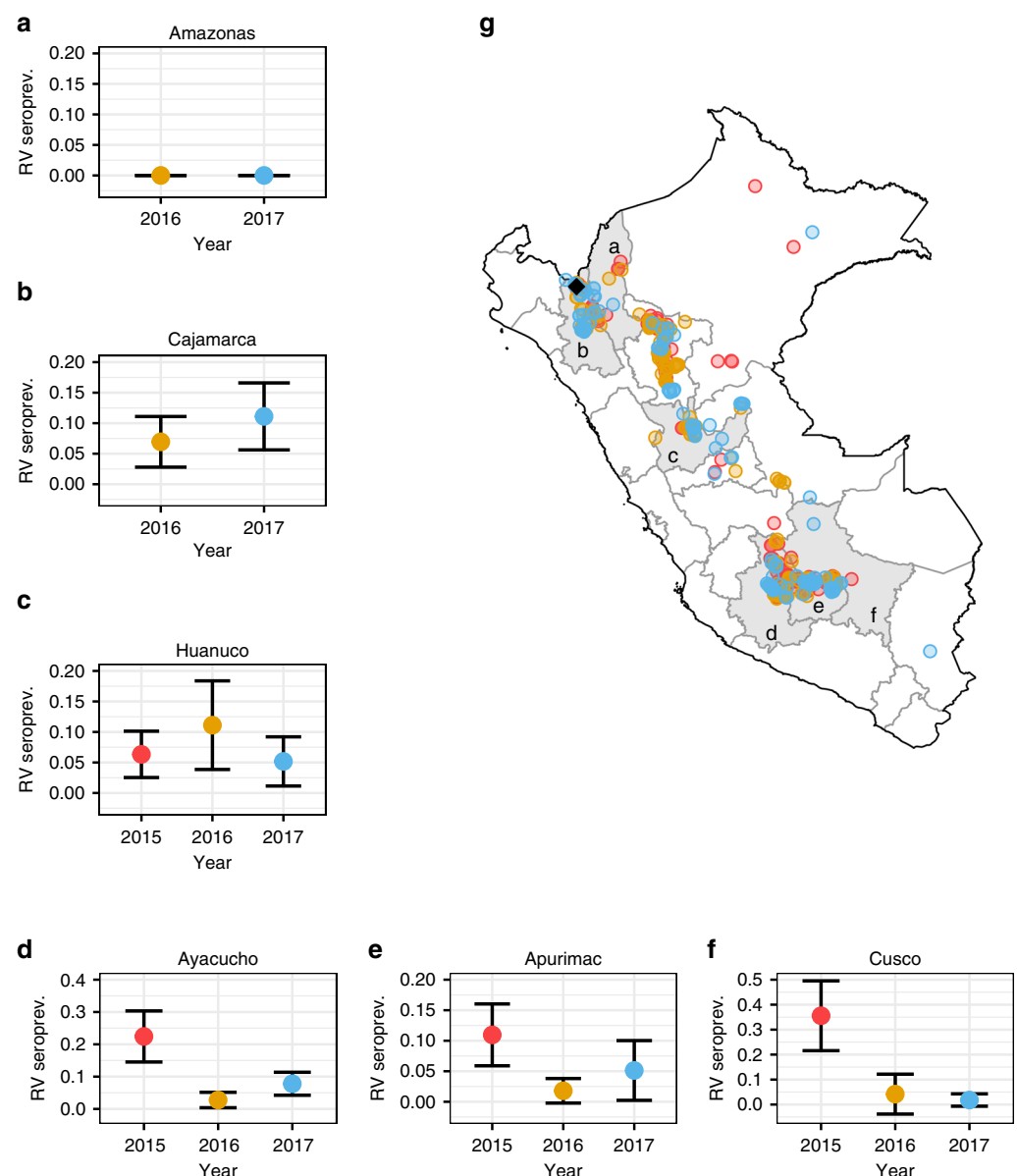

**Fig. 2 Seroprevalence of VBRV in D. rotundus and geographic distribution of outbreaks in cattle. a–f** The seroprevalence of VBRV in *D. rotundus*, grouped by year and department within Peru, are shown in the surrounding plots (**a–f**), with the location of each department marked by the corresponding letter on the map. Seroprevalence within a department is displayed as the mean average with 95% confidence intervals based upon sample size. Sample sizes for A-F in ascending year order are as follows: A-5, 21; B-144, 126; C-158, 72, 116; D-107, 181, 218; E-146, 167, 78; F-45, 24, 111. **g** Locations of laboratory confirmed rabies mortality in livestock between 2015 and 2017 are shown coloured by year (2015=red, 2016=yellow, 2017=blue) on the map of Peru. The location from which the PCR positive bat saliva sample originated is shown by the black diamond. Source data are provided as a Source Data file.

and amino acid sequences obtained from Peruvian bats, as well as sequences from other bat and non-bat BHVs on GenBank, was carried out to evaluate whether sympatric bat species shared common BHVs or maintained phylogenetically compartmentalised, host-associated viruses and to evaluate broader evolutionary patterns of co-speciation and host shifting among mammals. The nucleotide tree (Fig. 3, main image) revealed distinct, host-species associated clades in the bat species with multiple samples (*D. rotundus*, *Glossophaga soricina*, *Myotis oxyotus*). Those with only one sample per species (e.g. *Sturnira tildae*, *Saccopteryx bilineata*, *Rhynchonycteris naso*) also had distinct viruses that may also comprise host-associated clades. The strongly supported *D. rotundus* – associated clade (hereafter, DrBHV) contained sequences from vampire bat populations across Peru and no

vampire bat-derived samples occurred elsewhere in the phylogenetic tree. At even broader geographic levels, BHVs from bats in the genus *Myotis* clustered together, despite being sampled on different continents, with a single clade comprising of viruses from Peru and Spain, suggesting ancient circulation of BHVs in this genus. The exception to viral compartmentalisation by bat taxonomy was a sample taken from a single *Artibeus lituratus* bat (total sample size $n = 2$, positive=1) which clustered within DrBHV. Cytochrome B PCR and sequencing confirmed that this sample originated from an *A. lituratus* bat, suggesting a possible cross-species infection within the family *Phyllostomidae*. At deeper nodes, both bat BHVs as a whole and viruses from individual bat families (i.e. *Phyllostomidae* and *Vespertilionidae*) formed paraphyletic groups; however, these branches were poorly

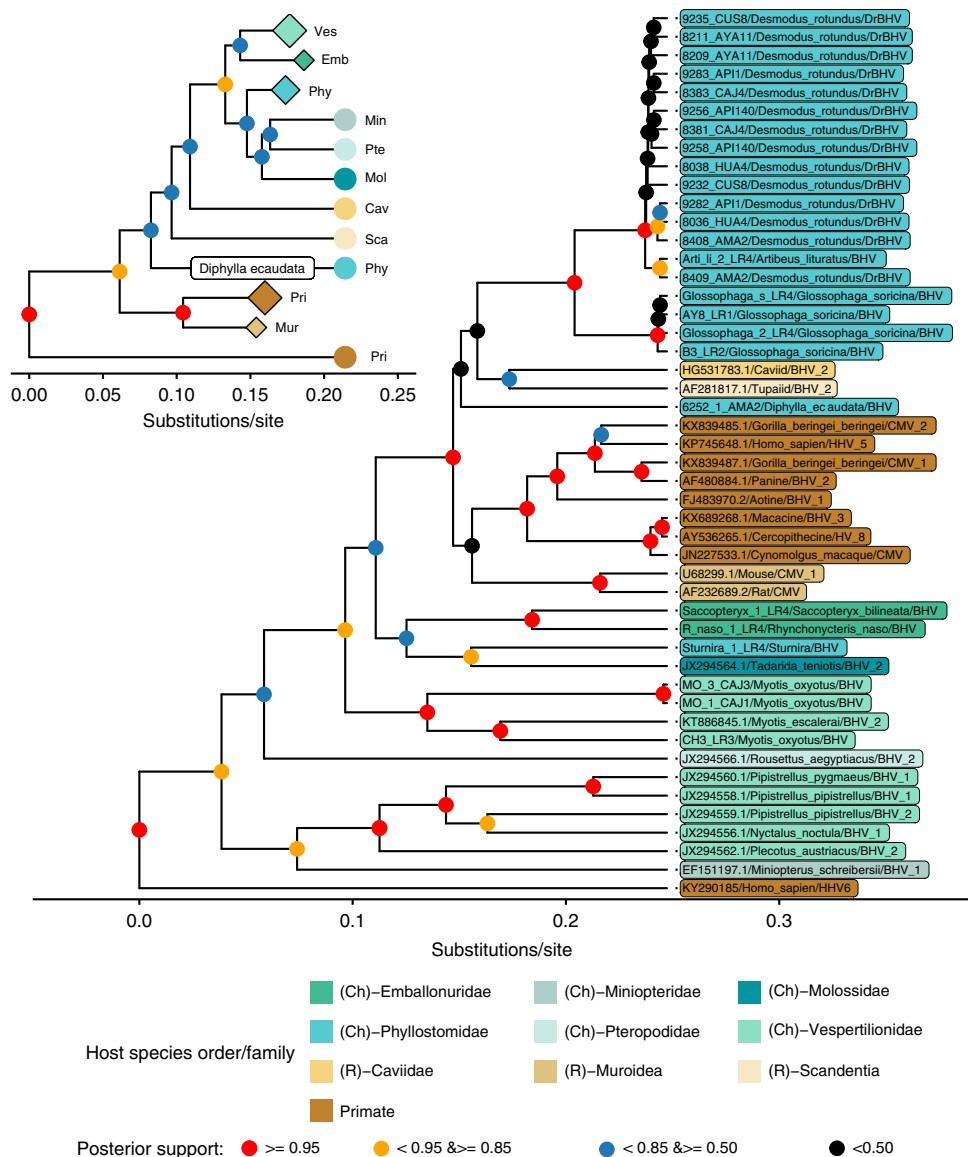

**Fig. 3 Bayesian phylogenetic tree of mammalian BHV shows clustering of virus diversity by host taxonomy.** Nucleotide-based phylogeny of UL89 sequences (304 bp) showing clusters of BHVs that were generally restricted to single species. Posterior probabilities are represented by node colour. Tip labels are coloured according to host taxonomy (family level for orders Chiroptera and Rodentia, and order level for Primates) and include the host species in which the BHV was detected and GenBank accession number where appropriate. The inset amino acid sequence-based phylogeny recovers the monophyly of bat BHVs and indicates bat family specific clustering of viral diversity, apart from a BHV from *Di. ecaudata* which groups separately from all other BHVs from Phyllostomid bats. Clades are collapsed into bat families, primate and rodent groups, represented by the diamond tips, proportional to the size of the collapsed clade. Individuals are represented by circular tips. Colours and family/order abbreviations correspond to those in the main figure. Both datasets include human herpesvirus 6A (HHV6, genus *Roseolovirus*) as an outgroup.

supported (posterior probabilities <0.85) relative to the >0.95 support for the host-species specific clades (Fig. 3). The phylogenetic analysis of the amino acid sequences reduced the distinction between closely related viruses, but largely recovered the monophyly of bat-associated BHVs and revealed bat-family associated viral clades (Fig. 3, inset). One exception was a BHV recovered from the hairy-legged vampire bat (*Diphylla ecaudata*) which grouped more closely with the tupaiid (common tree shrew, Malaysia) BHV than bat BHVs.

We next carried out a host-virus co-phylogenetic analysis to quantify support for co-speciation versus host shifting in the evolution of bat herpesviruses, and therefore the potential for future host shifts. The co-phylogenetic analysis, in which the virus phylogeny is scaled to fit the host phylogeny, calculates the global sum of squared residuals ($m^2_{xy}$) for the observed host-virus

network, and compares this to those of randomly generated interaction networks, producing a null distribution of $m^2_{xy}$ values. The analyses indicated significant congruence between the host species phylogeny and the maximum clade credibility trees of BHVs at both the (a) nucleotide ($m^2_{xy} = 3.535$, $P < 0.001$, null $m^2_{xy} = 6.671$, standard deviation [SD] = 0.358) and the (b) amino acid levels ($m^2_{xy} = 1.572$, $P < 0.001$, null $m^2_{xy} = 6.315$, SD = 0.355), where none of the 10,000 random networks showed more phylogenetic congruence than the observed network. Since variable posterior support at deeper nodes in BHV phylogenies indicated considerable uncertainty which could influence the co-evolutionary analyses, we repeated this analysis on 100 trees that were randomly sampled from the posterior distributions of each phylogenetic analysis, both of which showed congruence between host and virus trees (average values of test statistics over 100 trees:

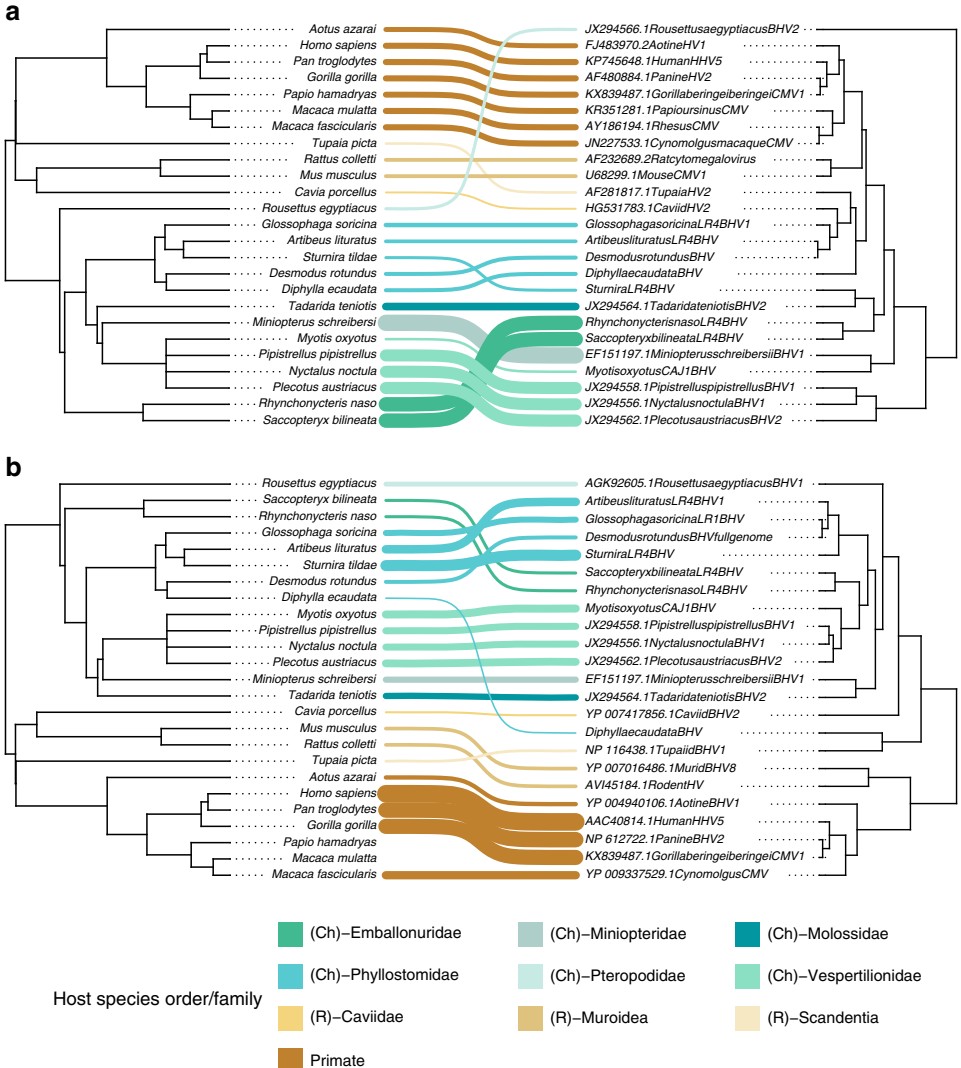

**Fig. 4 Co-phylogenetic analysis of hosts and betaherpesviruses reveal co-evolution across mammals.** Co-phylogeny of host species (phylogeny extracted from the mammalian super tree[71]) (left) with BHV, based on viral nucleotide (**a**) and amino acid (**b**) sequences. The lines between trees show links between virus and host phylogenies and are coloured by host family. Link weight is inversely proportional to the PACo squared residual for each species; heavier lines therefore reflect the likelihood that a pair represents a co-evolutionary link.

(a) $m^2_{xy} = 3.616$ SD $= 0.360$, $P < 0.001$; (b) $m^2_{xy} = 1.519$ SD $= 0.335$, $P < 0.001$). A jackknife procedure testing how each host-virus link contributed to the overall measure of congruence found that 61 and 56% of the host-virus links were congruent in the amino acid tree and nucleotide tree, respectively (95% confidence interval below the mean of all squared jackknife residuals; Fig. 4). Support for the primate and rodent links was much higher in the amino acid tree compared to the nucleotide tree (all link supports can be found in Supplementary Table 4), reflecting the separation of bat viruses into a monophyletic clade. The position of the BHV from *Diphylla ecaudata* was incongruent with the host phylogeny in both analyses, excluding co-divergence of this virus.

**DrBHV capacity for super-infection and genomic organisation.** To determine the presence of multiple strain infections within individuals, two PCR-positive *D. rotundus* saliva samples (IDs 10144 and 10148, see Supplementary Table 5 for details) were selected for shotgun metagenomic sequencing. Reads from 10148 were used to generate a consensus 231kbp consensus

genome. Re-mapping reads to the consensus genome revealed 4779 nucleotides (~2%) that had single nucleotide polymorphisms (SNPs) in which more than 10% of reads differed from the consensus. The distribution of SNPs varied across the genome (0.7–5.4% of positions, analysed in 1000 bp sections). Comparisons of short sections of the genome spanned by multiple unique reads revealed linked SNPs, and enumeration of such reads in areas of high coverage corroborated the presence of multiple unique DrBHV strains in both bats (Fig. 5). Due to the size of reads produced by shotgun sequencing (150 bp), and the low coverage depth (average 8.28 reads/base), it was only possible to use short windows (50 bp) for analysis; it was therefore difficult to quantify how many strains were present in each bat. Despite this, sample 10148_KF29 contained two major strains in each window tested, which differed by >10% at the nucleotide level (Fig. 5). Lower coverage in sample 10144_KF29 meant that distinguishing true strains from sequencing error was not always possible. However, two sequences which met our criteria for being considered distinct strains were observed in some locations (Fig. 5a, C), one of which was identical to a strain found in sample 10148_KF29 (Fig. 5c), supporting a multiple-strain infection.

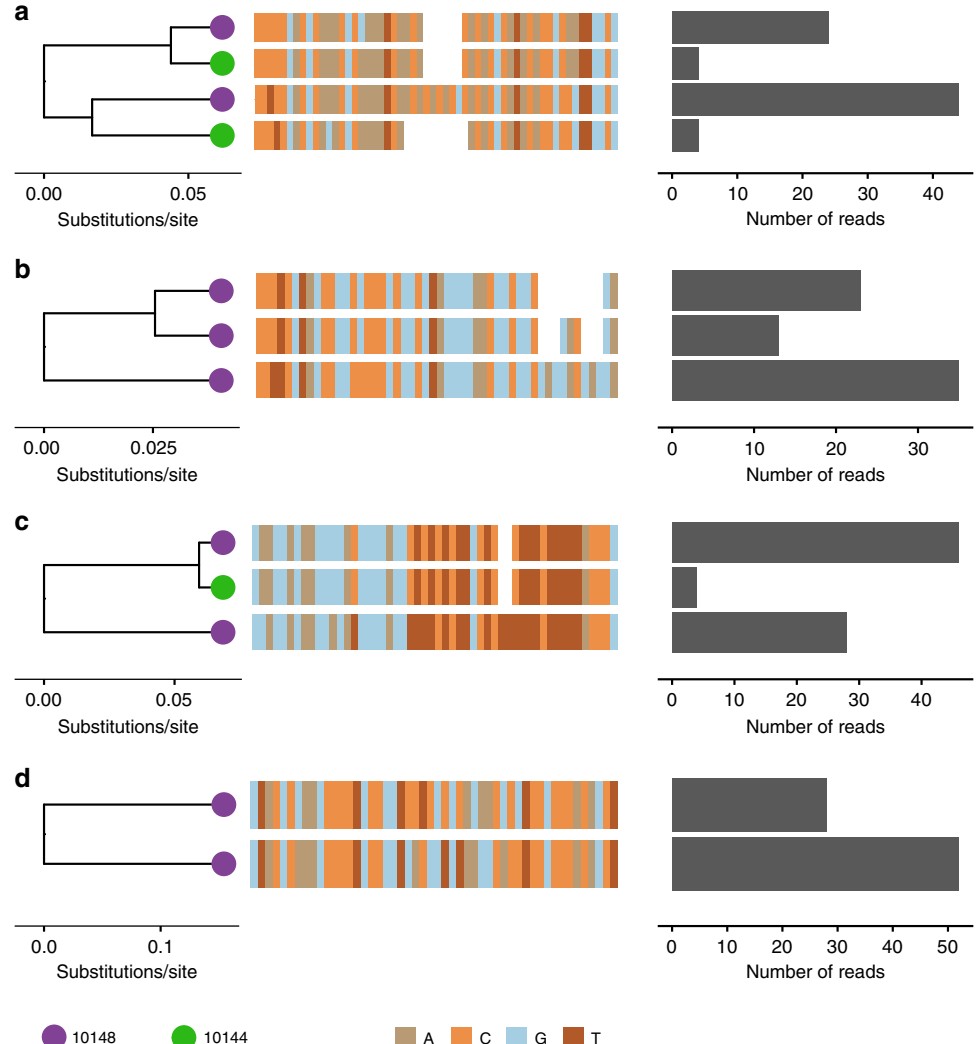

**Fig. 5 Metagenomic sequence data reveal multi-strain DrBHV infections in individual bats.** The phylogenies (left), nucleotide sequences (centre), and read count (right) of DrBHV strains present in the alignment of samples 10148_KF29 and 10144_KF29 (where read depth was sufficient for reads to be included) with the DrBHV consensus genome sequence. Nucleotide locations: (**a**) 47620– 47671 (equivalent to part of *Miniopterus schreibersii* BHV B7D8 B36), (**b**) 121057–121108 (B70), (**c**) 141567–141618 (B85) and (**d**) 229640–229691 (B161). These locations were selected due to high coverage depth in sample 10148_KF29. The tree tips are coloured by the individual bat from which the sequence originates.

Due to the presence of multiple strains within both samples, the consensus DrBHV genome sequence (reconstructed from the higher-coverage sample 10148; Supplementary Table 5) was chimeric. Nevertheless, this sequence showed a similar gene organisation and length to other BHVs (Supplementary Fig. 1). Genome annotation showed highest protein similarity by BLASTx to the bat *Miniopterus schreibersii* BHV B7D8 (223kbp) genome, the only available complete bat-derived BHV genome, although 24/114 predicted proteins had insufficient BLAST similarity to be assigned (ORFs predicted by GeneMarkS). Recombinant BHV vaccines have inserted genes into several locations along the vector genome, including UL82[40] (encoding for IE protein pp71) in HCMV, IE2[41] in MCMV, and RhCMV ORF Rh211[25,42,43] (US27), Rh107[43], Rh110[42], and Rh186-9[25] (US8-11). Whilst similar coding regions within DrBHV could not be identified for all of the above proteins, a homologue for UL82 in DrBHV could serve as a potential site for VBRV glycoprotein insertion.

## Discussion

Preventing zoonoses by controlling transmission within natural animal reservoirs offers an exciting complement to ongoing efforts to manage these infections after they enter human and domestic animal populations. This study shows how integrating information from metagenomic sequencing, field studies and viral genomics can begin to overcome the logistic hurdles to managing zoonoses within inaccessible wildlife populations. Utilising long running field studies in vampire bats, we have identified characteristics of a betaherpesvirus, DrBHV, that make it a promising candidate for further study as a transmissible vector for vaccine targeting vampire bat rabies.

Our co-phylogenetic analyses confirm and extend BHV co-evolution with mammals[44]. However, despite the pattern of strong host-association, we found two possible exceptions. First, we observed one *A. lituratus* fruit bat infected with the vampire bat-associated BHV. The lack of divergence of this virus from DrBHV most parsimoniously is explained by a cross-species

transmission event that had little or no onward transmission in *A. lituratus* at the time of our sampling or by a contaminating presence of DrBHV in the oral cavity of *A. lituratus* without actual infection (Fig. 1). Interestingly, rabies transmission has also been noted between these species and is potentially explained by their frequent ecological overlap together with their evolutionary relatedness within the family *Phyllostomidae*[45,46]. Given our low sample size for *A. lituratus*, further instances of cross-species transmission may be found, and this also holds true for other species of bats within *Phyllostomidae* that roost with *D. rotundus* and were poorly sampled, such as *G. soricina*. Additional sampling is required to determine the true frequency of BHV transmission among phyllostomid bat species. Second, we observed a BHV in *Di. ecaudata* that was paraphyletic to all other bat BHVs. Given that this species feeds on the blood of birds and wild mammals, we suspect the unexpected evolutionary placement of this virus may be a result of dietary contamination acquired from infected prey. Additional testing of *Di. ecaudata*, particularly using blood samples, would be helpful to exclude active infection with this virus and to evaluate whether *Di. ecaudata* maintains other BHVs that group within the larger bat clade, but were missed due to our limited sampling of this species ($N = 2$). Aside from these exceptions, BHVs formed taxon-associated clusters within bats, consistent with host specificity. Given the expected host specificity of BHVs[20,47] and the epidemiological and co-evolutionary results shown here, the available evidence most strongly argues that DrBHV-vectored vaccines would be unlikely to infect mammals outside of the *Phyllostomidae* family of New World bats and may only rarely spread beyond *D. rotundus*.

We observed a high prevalence of BHV infection in vampire bats relative to other bat species. This pattern may be explained by the high levels of social grooming within this species as well as food sharing through regurgitation of blood, both of which could facilitate BHV transmission in saliva[48]. Since the prevalence of a wild-type virus reflects the hypothetical maximum coverage that it would achieve as a vector, the prevalence of DrBHV in vampire bats implies that a rabies vaccine vector based on this virus might infect and vaccinate nearly all individuals. Nevertheless, we emphasise that this estimate represents an upper bound since the addition of a target virus gene into DrBHV may diminish transmissibility. Moreover, our metagenomic sequencing revealed that multiple BHV strains likely co-circulate in vampire bat populations. Strain-specific prevalence may therefore be the most appropriate proxy for the maximum achievable vaccine coverage. Although our PCR was not suited to defining evolutionarily independent strains within vampire bats, future studies using strain-specific diagnostics are needed to estimate the frequency, intensity and geographic scope of vaccination that would be necessary to control or eliminate VBRV.

Importantly, lower efficiency spread than implied by our global prevalence estimates may actually be desirable. From a safety standpoint, transmissible vaccines should have the minimum transmissibility to meet management needs and even weakly transmissible vaccines are predicted to be surprisingly effective in disease control[10,17]. Indeed, the more ambitious aim of eradication only requires greater transmissibility of the vaccine than the target zoonosis ($R_{0,Vaccine} > R_{0,Target}$). For VBRV, persistence relies on metapopulation structure and $R_0$ within single vampire bat colonies is estimated to be $< 1$[7]. Consistent with other field studies, our serological and RT-PCR findings that a sizable minority of bats are exposed to rabies virus, but few succumb to productive infections, further supports the low transmissibility of VBRV[8]. Moreover, bats with antibodies acquired from natural VBRV exposures are likely to have a degree of protective immunity[7,49], further lowering the transmissibility requirements for a DrBHV-vectored rabies vaccine. Although the $R_0$ of DrBHV

cannot be calculated from our data, infection of nearly all vampire bats suggests its transmissibility likely exceeds that of VBRV. Our results therefore imply that a significant reduction of VBRV might in principle be achievable with a DrBHV vectored rabies vaccine with similar properties to the wild-type viruses that we studied.

Persistent, life-long infections of vampire bats by DrBHV were expected given the well-established ability of other BHVs to undergo phases of latency, in which viral load is low and no shedding occurs, and reactivation, in which viral load increases and shedding resumes[25]. Indeed, several lines of evidence shown here support persistent infections. First, acute infections would not be expected to have the consistently high prevalence across space and time that we observed for DrBHV. Second, although we were unable to conclusively rule out sequential re-infections, we found that 3/3 bats sampled multiple times between 2015 and 2018 remained infected up to four years after initially being BHV-positive. Third, we observed bats with low levels of DrBHV in saliva (as indicated by lack of detectable virus DNA in round 1 of the PCR), which we hypothesise may have been sampled during the latent phase of infection[50]. If latency and reactivation are preserved in the vaccine vector, reactivations might naturally boost immune responses to VBRV over the course of the vaccinated bat's lifetime, while potentially providing additional opportunities for vaccine transmission.

Capacity for multiple-strain infections is a key prerequisite for virally vectored transmissible vaccines, since immune responses to wild-type versions of the vector might inhibit vaccine spread[18]. In both individuals tested, our metagenomic data showed multiple sequences corresponding to at least two distinct strains, indicative of naturally occurring super-infections. Co-transmission of multiple variants is unlikely to be the sole explanation given the well-documented capacity of BHVs to super-infect. For example, in cytomegaloviruses and muromegaloviruses, down-regulation of MHC-class 1 on infected cells prevents communication of infection to T-cells, allowing frequent super-infections despite the large and lifelong T-cell and antibody response elicited by primary infections[22,51,52]. Within-host evolution occurs in persistent HCMV infections[53] and may also explain some of the DrBHV polymorphisms we observed separating very closely related strains. However, given the slow evolutionary rate of BHVs, which is comparable to that of other DNA viruses, strain divergences of up to 15% in single bats can only be explained by super-infection or transmissions of multiple, evolutionarily distinct strains. The presence of both closely related and distantly related strains in a single bat suggests both intra-host evolution and super-infection (Supplementary Fig. 2). Beyond superinfection of divergent strains, an ideal vector would also re-infect individuals previously infected with the same strain of virus. This is because the inevitable loss of recombinant material from the vaccine may create viable mutants that compete with the intact vaccine strain. The existence of multiple variants of DrBHV (Fig. 5), raises the possibility that several strains circulate around Peru, each with its own prevalence and geographic range. Population genetics of both vampire bats and VBRV support this prediction, showing major barriers to gene flow among geographical regions of Peru[37,54]. Therefore, DrBHV strains that are locally prevalent in one area could be engineered and released in other areas where they do not self-compete, circumventing competition with wild-type viruses of the same strain as the vaccine. Understanding whether mutant vaccine derived strains may frustrate longer term vaccine applications requires same-strain reinfection experiments in captive bats. Longitudinal monitoring of vaccine-infected individuals is also needed to quantify the dynamics of latency, reactivation and super-infection[55,56].

Numerous obstacles must be resolved prior to deploying DrBHV as a transmissible rabies vaccine. Although our data are consistent with transmission in saliva, detection of viral DNA in blood suggests systemic infection and the possibility of transmission through additional bodily fluids, or transplacental transmission, as can occur for HCMV[57]. The geographic structure of DrBHV populations would also determine the spatial distribution of vaccines, in regards to both initial deployment, and predicted range and timescale of spread. Phylogenetic studies of different strains over time could be one way to better understand spatiotemporal spread. Safety and regulatory issues must also be carefully considered. For example, WHO guidelines for recombinant dog rabies vaccines discourage those with potential to be shed and spread infectiously (http://www.who.int/rabies/resources/Oral_Vaccination_of_Dogs_Against_Rabies/en/). Although these guidelines were based on the presumption that vaccines would be attenuated forms of pathogenic viruses, additional safety testing, perhaps in conjunction with a modernisation of the regulatory environment, will be required prior to field trials of DrBHV-vectored rabies vaccines. Advances in deliberately transmissible therapies in other areas of disease control, ranging from early stages of concept development (e.g. use of genetically heritable CRISPR-edited genes to generate Lyme disease-resistant mice[58]), to limited experimental trails (e.g., immunocontraception[59]), to large-scale applications (e.g., release of *Wolbachia*-infected mosquitoes to combat dengue virus infections), and provide frameworks to guide how new technologies can be applied rationally. Finally, we emphasise that our work involved partial sequencing of wild-type viruses, which were used as a proxy for a still-hypothetical vaccine. A vital next step would be to isolate DrBHV strains so that their biological properties (such as capacity for latency, re-activation and super-infection) can be confirmed in vitro and in vivo and so that a vaccine can be engineered and confirmed to retain these desirable properties.

In summary, we have identified a BHV in common vampire bats (DrBHV) that meets the ecological, epidemiological and virological prerequisites of an ideal transmissible vaccine vector. Given that VBRV is naturally prone to extinction and that our results suggest efficient DrBHV transmission across all demographic groups and populations of vampire bats tested, a protective DrBHV-vectored vaccine that transmits similarly to or even less efficiently than wild-type DrBHV could conceivably eliminate VBRV circulation. Importantly, VBRV and most other bat-associated rabies viruses are maintained in species-specific transmission cycles without alternative hosts that could otherwise compromise vaccination campaigns[37,60]. Moreover, rabies vaccines are broadly protective across *Rabies lyssavirus* strains and even protect against other lyssaviruses, suggesting that viral antigenic evolution would not be a relevant barrier[61,62]. Finally, decades of experience in vampire bat population control have equipped most Latin American countries with the operational capacity to implement vaccination campaigns alongside or in lieu of culling efforts. Although BHVs may not be suitable vectors for all bat-associated viruses, we illustrate that combining viral metagenomics, field studies and co-evolutionary approaches can identify promising candidate vectors. This approach could be applied to any bat species for which reservoir-targeted disease control is desirable to prevent zoonotic spillover. We encourage serious consideration of the potential for recombinant virally vectored vaccines to control zoonotic bat viruses within their reservoir hosts.

## Methods

**Sample collection.** Bats were captured between 2015 and 2018 at 37 sites across eight administrative regions (Amazonas, Apurímac, Ayacucho, Cajamarca, Cusco, Huánuco, Lima and Loreto) of Peru. Bats were captured using mist nets and harp traps, then placed in individual cloth bags before processing and sampling[29]. Bats were aged by observation of the epiphyseal–diaphyseal fusion[63]; sex and reproductive status were also recorded. Saliva swab samples were collected by allowing bats to chew on sterile cotton-tipped wooden swabs (Fisherbrand) for 10 s. Whole blood samples were also collected on swabs after puncturing the propatagial vein with a sterile 23-gauge needle. All swabs were stored in either 1 ml RNALater (Ambion), virus transport medium (VTM, phosphate-buffered saline supplemented with 10% foetal calf serum, and double-strength antibiotic/antimycotic [200 U/ml penicillin, 200 g/ml streptomycin, and 0.5 g/ml fungizone amphotericin B]) or phosphate-buffered saline (PBS), and then stored at −80 °C until further analyses. All capture and sampling of bats were approved by the Research Ethics Committee of the University of Glasgow School of Medical, Veterinary and Life Sciences (Ref081/15) and by the University of Georgia Animal Care and Use Committee (A2014 04-016-Y3-A5). Field collections were authorised by the Peruvian government (RD-009-2015-SERFOR-DGGSPFFS, RD-264-2015-SERFOR-DGGSPFFS, RD-142-2015-SERFOR-DGGSPFFS, RD-054-2016-SERFOR-DGGSPFFS). Livestock sample data were published previously (see data availability).

**Nucleic acid extraction.** Nucleic acid extractions from swabs were performed on a Kingfisher Flex 96 automated extraction instrument (ThermoFisher Scientific) with the BioSprint One-For-All Vet Kit (Qiagen) using a modified version of the manufacturer's protocol for purifying viral nucleic acids from swabs[29].

**Amplification and Sanger sequencing of betaherpesviruses.** Nucleic acid extractions were analysed using a semi-nested PCR with primers that amplify a region of the highly conserved terminase gene UL89: BHV-8F: 5′-TTC ATC TCG TCC ACC AAC AC-3′ (round 1 and 2 forward primer), BHV-7R: 5′-TGT AGC GGA ACA CGT CGA AC-3′ (round 1 reverse primer) and BHV-8R: 5′-CGA TGG TCT CGT CCA TGA AG-3′ (round 2 reverse primer). Primers were adapted to be BHV-specific from those used in Pozo et al.[30]. The PCR protocol was as follows: 94 °C for 2 min, 40 cycles of 94 °C for 30 s, 40 °C(round 1)/35 °C(round 2) for 3 min, 72 °C for 30 s, and a final 72 °C for 5 min. HCMV was used as a positive control. The amplification of BHV DNA was confirmed by gel electrophoresis (2% agarose in Tris-acetate-EDTA buffer) examined for the presence of both round 1 (~315 bp) and round 2 (~219 bp) bands of the expected size. A selection of positive round 1 and round 2 samples (all non-*D. rotundus* samples and ~10% of *D. rotundus* saliva samples, numbers available in Supplementary Table 2) were then purified using the QIAquick PCR purification kit (Qiagen), and Sanger sequenced in both directions by Eurofins Genomics. Forward and reverse sequences, trimmed to remove primers and low confidence bases, were assembled to create a 304 bp consensus sequence for each sample using CLC Genomics Workbench v7.5 (https://digitalinsights.qiagen.com).

**Molecular confirmation of bat species identities.** Bat host species was confirmed by sequencing a 450 bp cytochrome B sequence using the primers and PCR protocol from Martins et al.[64]. Briefly, Bat 05 A 5′-CGACTAATGACATGAAAAAT CACCGTTG-3′; Bat 04 A 5′-GTAGCTCCTCAGAATGATATTTGTCCTC-3′ primers; 5 min at 94 °C, 35 cycles of 30 s at 94 °C, 45 °C for 45 s, 72 °C for 70 s and a final step of 72 °C for 10 min. The samples tested were the *A. lituratus*, *Di. ecaudata* and *S. lilium* saliva samples.

**Amplification and sequencing of vampire bat rabies virus.** 123 of the 128 *D. rotundus* saliva samples that were tested for DrBHV were also tested by nested RT-PCR for the presence of rabies virus. The five samples that were not tested had insufficient nucleic acid extract for testing. The RT-PCR was carried as described by Kuzmin et al.[65] using primers found in Supplementary Table 6: RT step of 90 min at 42 °C, 40 PCR cycles of 30 s at 94 °C, 30 s at 37 °C, 90 s at 72 °C and a final 10 min at 72 °C.

**Detection of rabies virus neutralising antibodies.** To detect and quantify rabies virus neutralising antibodies (RVNA), we used a pseudotype micro-neutralisation rapid fluorescent focus inhibition test[38]. This test uses a combination of microscopic imaging of cellular infectivity and generalised linear mixed modelling (GLMM) to produce predicted RVNA titres in units of International Units/ml. Samples were classified as seropositive or seronegative using a threshold of 0.166 IU/ml, which was previously shown to balance sensitivity and specificity most appropriately[38]. GLMMs were fit using the lme4 package in R[66]. This yielded a similar global seroprevalence to that observed in a previous study of Peruvian vampire bats using the gold standard Rapid Fluorescence Focus Inhibition Test (RFFIT)[8].

**Statistical analysis.** Prevalence by species was assessed by using a point estimate of the proportion of bats infected, with binomial confidence intervals (R (R studio v1.1.456), binom package, method = bayes). A binomial generalised linear mixed model (GLMM) followed by post-hoc pairwise comparisons of estimated prevalence by species was used to test the significance of differences in prevalence.

Within vampire bats ($N = 128$), the statistical significance of factors that may influence DrBHV infection was tested, including age (adult, sub-adult or juvenile),

sex (male or female), eco-region (Amazon, Andes or Coastal), year, season and roost structure (single versus mixed-species roost). Statistical modelling used a binomial GLMM employing the lme4 package of R[66], with the site from which the bat was captured treated as a random effect.

**Phylogenetic inference**. A dataset was constructed for phylogenetic analysis, containing our own sequenced, and previously sequenced bat BHV sequences, and containing BHV sequences from other mammalian species, including apes, monkeys, tree shrews and rodents. These sequences were acquired by conducting a BLASTn search in GenBank (http://blast.ncbi.nlm.nih.gov/) using the DrBHV sequenced samples and selecting a representative viral sequence from a range of host species. The dataset included the relevant BHV human herpesvirus 6 A (genus *Roseolovirus*, https://www.ncbi.nlm.nih.gov/nuccore/KY290185) sequence for use as an outgroup, and can be found in full in Supplementary Table 7.

Consensus sequences from Sanger sequencing and those from GenBank were aligned using MUSCLE (EMBL-EBI). The most likely evolutionary model (GTR + gamma+invariant) was identified using jModelTest2 using the BIC criterion[67]. Phylogenetic reconstruction was carried out using BEAST v1.10.4 and associated software[68]. Markov chain Monte Carlo sampling of trees and parameters was run for 10 million generations, sampling every 1000 generations. Posterior traces were checked in Tracer v1.7.1 to evaluate convergence and select burn-in periods (10% was selected). Consensus trees were constructed using TreeAnnotator and visualised in FigTree v1.4.4 and R package ggtree[69]. Nucleotide sequences were also converted to amino acid sequences and the phylogeny reconstructed using Blosum62 + G + I in BEAST.

Co-phylogenetic analysis was used to identify the frequency of host switching and co-divergence by employing the package PACo in R[70]. PACo returns a residual sum of squares as an analysis of the Procrustean fit of the parasite (virus) phylogeny to that of the host, as a measure of congruence between the two phylogenies. Statistical significance was tested via a permutation test wherein the host-parasite association matrix (each host and parasite matrix were first normalised between 0 and 1 due to the differing scales of the two phylogenies) was randomised and the residual sum of squares was calculated for each permutation. The consensus BHV nucleotide- and amino acid-based trees, and 100 trees randomly selected from the BEAST posterior distribution (excluding the burn-in), were tested using this goodness-of-fit test for significance with 10,000 permutations for the consensus tree and 1000 permutations each for the posterior trees, the individual *p*-values from which were then averaged. This allowed assessment of co-evolutionary relationships whilst accounting for uncertainty in the viral evolutionary history. A jackknife procedure was used to estimate the squared residual and its 95% confidence interval for each individual link. Host trees were extracted from the mammalian super-tree[71] in R using ape[72]. Ape was also used to construct co-phylogenetic linkage plots of the consensus trees.

**Whole genome sequencing and bioinformatics analysis**. Two of the *D. rotundus* samples that tested positive for BHV were selected for whole genome sequencing. One sample (10148) was selected due to its strong PCR positive band on gel electrophoresis. The second sample (10144) had a weaker band from PCR. Together, these samples should be representative of different levels of infection by DrBHV. DNA was extracted from each saliva swab sample both with and without DNase treatment in order to remove host DNA, which can reduce the number of host and bacterial reads and enrich viral reads in the final pool[29]. Sequencing libraries were prepared using the DNA Library Preparation Kit for Illumina (KAPA Biosystems) with index primers from New England Biolabs (NEB), and sequenced on the Illumina NextSeq platform to produce approximately 40 million paired-end reads of 150 bp (Supplementary Table 5).

Reads with significant BLAST similarity to the vampire bat genome, as well as low-quality reads, were removed, and remaining reads were trimmed to remove adaptors using trim_galore and diamond v0.9.25 as part of the allmond bioinformatics pipeline for viral discovery (https://github.com/rjorton/Allmond). Reads were then assembled into contigs using SPAdes v3.10.1 (de novo assembly), Bowtie2 v2.3.5.1 and BWA v0.7.17 and samtools v1.9 (reference assembly). Contigs were reference aligned to other BHV genomes retrieved from GenBank (HHV5, NC_006273 (https://www.ncbi.nlm.nih.gov/nuccore/NC_006273); panine BHV 2, AF480884(https://www.ncbi.nlm.nih.gov/nuccore/AF480884); murid BHV 1 NC_004065 (https://www.ncbi.nlm.nih.gov/nuccore/NC_004065); bat BHV B7D8, JQ805139 (https://www.ncbi.nlm.nih.gov/nuccore/JQ805139)). A manual search for sequence overlaps between contigs was then used to produce a nearly continuous DrBHV genome sequence. Cleaned and filtered reads were subsequently realigned to this sequence using Bowtie2 to increase coverage across the genome. BLASTx was used to annotate the consensus genome sequence with homologous genes from other BHVs. Open reading frames and corresponding proteins were predicted using GeneMarkS[73]. The resulting proteins were compared to those found in other BHVs (BLASTp), and those with a match were used to annotate the genome using CLC Genomics Workbench v7.5.1 (https://digitalinsights.qiagen.com). The raw reads were then re-aligned to the final consensus DrBHV genome sequence in order to search for single nucleotide polymorphisms (SNPs) that might indicate super-infection, using samtools mpileup SNP calling (Supplementary Data 2). Only nucleotides with a coverage of >10 reads were considered. In Fig. 5, phyloscanner was used to search for the

different sequences present in the 10148_KF29 and 10144_KF29 aligned reads, in 50 bp windows. This window size was selected to maximise the number of reads that could be included, as a read must span the full window. Extracted reads were visualised using R package ggtree.

**Reporting summary**. Further information on research design is available in the Nature Research Reporting Summary linked to this article.

## Data availability

Results of the betaherpesvirus PCR, rabies virus RT-PCR and rabies virus neutralising antibody test, for Peruvian bats, generated and/or analysed during the current study are available in the figshare repository https://doi.org/10.6084/m9.figshare.13090214. UL89 partial sequences used for phylogenies have been uploaded to GenBank, with the following accession numbers: MT432305-16 and MT912480-93 (https://www.ncbi.nlm. nih.gov/nuccore/MT432305, https://www.ncbi.nlm.nih.gov/nuccore/MT432306, https:// www.ncbi.nlm.nih.gov/nuccore/MT432307, https://www.ncbi.nlm.nih.gov/nuccore/ MT432308, https://www.ncbi.nlm.nih.gov/nuccore/MT432309, https://www.ncbi.nlm. nih.gov/nuccore/MT4323010, https://www.ncbi.nlm.nih.gov/nuccore/MT4323011, https://www.ncbi.nlm.nih.gov/nuccore/MT4323012, https://www.ncbi.nlm.nih.gov/ nuccore/MT4323013, https://www.ncbi.nlm.nih.gov/nuccore/MT4323014, https://www. ncbi.nlm.nih.gov/nuccore/MT4323015, https://www.ncbi.nlm.nih.gov/nuccore/ MT4323016, https://www.ncbi.nlm.nih.gov/nuccore/MT912480, https://www.ncbi.nlm. nih.gov/nuccore/MT912481, https://www.ncbi.nlm.nih.gov/nuccore/MT912482, https:// www.ncbi.nlm.nih.gov/nuccore/MT912483, https://www.ncbi.nlm.nih.gov/nuccore/ MT912484, https://www.ncbi.nlm.nih.gov/nuccore/MT912485, https://www.ncbi.nlm. nih.gov/nuccore/MT912486, https://www.ncbi.nlm.nih.gov/nuccore/MT912487, https:// www.ncbi.nlm.nih.gov/nuccore/MT912488, https://www.ncbi.nlm.nih.gov/nuccore/ MT912489, https://www.ncbi.nlm.nih.gov/nuccore/MT912490, https://www.ncbi.nlm. nih.gov/nuccore/MT912491, https://www.ncbi.nlm.nih.gov/nuccore/MT912492, https:// www.ncbi.nlm.nih.gov/nuccore/MT912493). The VBRV sequence from bat 6024 has the accession number MT891038. The chimeric consensus sequence and aligned sequence reads have been submitted to the SRA with bioproject ID: PRJNA631425, and run IDs, SRR11789719-20 (https://trace.ncbi.nlm.nih.gov/Traces/sra/?run=SRR11789719, https:// trace.ncbi.nlm.nih.gov/Traces/sra/?run=SRR11789720). Source data are provided with this paper.

## Code availability

No unpublished/custom code was used to generate results presented here.

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

## Acknowledgements

M.G. was supported by a Medical Research Council scholarship via the MRC-CVR PhD programme (MC_UU_12014/12). A.d.S.F. was supported by the Medical Research Council (MC_UU_12014/12). A.J.D. was supported by the Medical Research Council (MC_UU_12014/3). L.B. and D.M. were supported by the Human Frontier Science Program (RGP0013/2018). D.J.B. was supported by an NSF Graduate Research Fellowship, NSF DEB-1601052, the ARCS Foundation, and the Explorer's Club. D.S. was supported by a Sir Henry Dale Fellowship, jointly funded by the Wellcome Trust and Royal Society (102507/Z/13/Z) and a Wellcome Senior Research Fellowship (217221/Z/19/Z). We thank Carly Trille for contributing to primer design and testing, Dan Haydon for discussions of early results, and three anonymous reviewers whose suggestions improved the manuscript.

## Author contributions

M.G. performed the formal analysis and investigation, including PCRs, and metagenomic sequencing preparation, and wrote the manuscript. A.B. performed sample extractions and PCRs and provided methodology and resources. L.B. performed extractions and developed the metagenomic sequencing protocol. D.M. performed rabies virus serology testing and subsequent analysis. A.D. assisted with genome assembly and interpretation. A.d.S.F. provided support and resources for metagenomic sequencing. C.T. and D.B. provided bat samples. D.S. conceptualised, managed and supervised the project, acquired funding and led review and editing of the manuscript, assisted by all other authors.

## Competing interests

The authors declare no competing interests.
