## [Peer Review File · Nature Communications]

Reviewers' Comments:

Reviewer #1:

Remarks to the Author:

In this paper, the authors report results from genetic surveys and phylogenetic analyses of a Betaherpesvirus, DrBHV, being investigated as a possible vector for a transmissible vaccine targeting rabies in vampire bats. In addition to presenting results evaluating the scope for DrBHV to serve as a useful vaccine vector for this specific application, the paper provides a template for a general approach to screening candidate vaccine vectors in other wildlife species. Primary results from the authors' analyses are that: 1) the prevalence of DrBHV is high within vampire bats, 2) multi-strain infections occur, and 3) host switching appears to be rare. These results are promising for the use of DrBHV as a vector for a transmissible vaccine as the authors argue. At the same time, as the authors recognize, this represents only an initial screen for suitability as a vector, not conclusive evidence that DrBHV will be optimal, or even suitable. Overall, this is an extremely clear paper that presents interesting results in a thoughtful and well-balanced way. My only substantive concern is the absence of information on rabies within these samples. In following paragraphs, I detail why I think this matters and also identify a few other issues that could be clarified or changed to improve the paper.

The focus of the work is identification of a vector suitable for developing a transmissible vaccine targeting rabies in vampire bats. Results are reported on the vector, but no information is provided on the incidence of rabies within the 128 vampire bats sampled. Given the authors' interest in rabies and background studying it in vampire bats, it seems likely this information was also collected from these samples and might be available. If so, the paper would be much stronger and the results easier to interpret if this information were provided. It matters because: 1) The presence of rabies in the same samples would demonstrate the distribution of DrBHV as measured here is relevant for rabies control, 2) The pattern of rabies infection relative to DrBHV infection would shed light on whether both viruses follow similar infection pathways, and 3) The prevalence of both viruses could be compared in a meaningful way, allowing for more concrete predictions about the likely performance of a transmissible vaccine targeting rabies using DrBHV.

Lines 76-78: "Furthermore, if vaccine-induced mortality were to occur in the reservoir host, this would not only pose animal welfare concerns, but also would have the undesirable effect of reducing vaccine-induced immunity at the population level." This may not be as relevant as it sounds because death also removes individuals from the susceptible pool which curtails the spread of the infectious disease, at least if transmission is density dependent.

I agree that the existence of multiple strains within a bat suggests reinfection. Unfortunately, for recombinant vector transmissible vaccine what really matters is whether reinfection with an identical strain is possible. This matters for two reasons. First, in most cases, the vector would be a strain already present within the target population and so the vaccine will compete with its progenitor unless it can re infect/superinfect. It might be possible to get around this issue by using a rare or infrequent strain as a backbone, but this comes at the risk of choosing a strain that is rare because it is not terribly transmissible/competitive in the first place. Second, no matter how good the engineering, the vaccine will almost certainly evolve to lose or downregulate its immunogenic cargo. Once it does, intact vaccine must compete with these mutants unless reinfection/superinfection by identical strains is possible. I think these more subtle issues need to be better developed to place the authors' very cool (but not comprehensive) results on reinfection/superinfection in their full context.

Reviewer #2:

Remarks to the Author:

Summary:

The objective of this study was to identify a non-pathogenic virus as a candidate for use in designing a transmissible vaccine against vampire bat rabies virus. Specifically, the study evaluates the suitability of a novel betaherpesvirus (BHV) in vampire bats (*D. rotundus*), by assessing BHV prevalence, host specificity, and co-evolution across a number of bat species and other mammalian hosts (i.e., rodents and primates). The authors conclude that a BHV strain in vampire bats (DrBHV) is a good candidate for a viral vaccine, due to their findings that it is found at high prevalence and the evolutionary history of BHVs suggest host-specificity. While the idea of using virus as a vector for wildlife vaccination is certainly cutting-edge and exciting, after reading the article, I found that this spin was slightly misleading with regards to the actual data, analyses, and results. Though the vaccine angle does provide justification for this work. Further detailed comments and suggested edits may be found below.

Major Comments:

>While, I do think the analyses are valid and provide some interesting information, the data are not explicitly testing the use of DrBHV as a potential vaccine vector (as the title and initial introduction imply), but rather describe the prevalence and distribution of a virus in bats where little previous information is currently available. I can see how the vaccine angle is justification for the study and should be emphasized somewhere in the paper, but think the authors may be overstating the potential for this (at this time) as a natural vector for a bat rabies vaccine.

>Along these same lines, the authors' claim the prevalence and host-specificity of BHV is evidence that the virus would be a suitable vector. But, the data seem to suggest a long evolutionary history of the virus with the host and it is not clear what the transmission rate would be (i.e. how long did it take for the natural strains of BHV to become established). How would an introduced recombinant virus achieve the same level of distribution and prevalence? There is an assumption here that emergence and persistence will be likely, but the time frame of the natural emergence process is unknown. And, their data also suggest multiple co-circulating BHV strains (or super-infections), so it is possible there are several viral strains with variable distributions and prevalence.

>L162-163: I'm not sure if there are enough samples per species here to really evaluate host specificity and degree of host shifts at the species level. The statement that there are host species-specific clades is not entirely supported with the fact that *A. lituratus* clusters with *D. rotundus*, though this is mentioned later in the paragraph. Could more convincingly support family-level clustering and higher level taxonomic co-evolution. But, again, I'm not convinced sampling is comprehensive enough to detect the variability in BHVs that may be present across bat host species. Do some sequences in the nucleotide-based phylogeny represent multiple samples? If so, this could be more clearly presented somewhere.

>L264-266: As commented on above, the time frame for establishing high prevalence of a transmissible vaccine may be quite long. While it is not feasible with these data, perhaps it would be possible to do a coalescent analysis (with genome-level BHV sequence data from vampire bats) to estimate the time to the most recent ancestor, in order to predict the time it would take for a vaccine developed from this virus to reach similarly high prevalence. Though, given the release parameters (number and distribution released), this may be paired with a modeling exercise or focused on a specific population as a case study. Again, these are just suggestions for future work, as I realize this is not the aim of the current paper.

Minor Comments and Suggested Revisions:

>Paragraph starting L143: In the text, where are the results shown in Fig. 1 comparing infection prevalence across species?

>L152: How many species are represented in the BHV prevalence value from blood samples? Is this just *D. rotundus* or other bat species as well?

>L153-154: I think this is an interesting discussion point. Does this mean that ~55% of positive samples (from saliva samples) could be due to environmental contamination? With a possible systemic infection prevalence as low as 45%, would this be sufficient for a successful vaccination campaign? Typically larger vaccination coverage is needed to reduce transmission, and for herd immunity.

>L186-7, 191: Is "P=0" correct, or should it be written as "P<0.0001"?

>L204: By "different," do you mean "multiple unique strains" in both samples?

>L212-213: Should this be 2-3 major strains at each location (for sample 10148)? Panel B appears to indicate 3 strains, or are you considering the top cluster as a single major strain?

>L232-238: The ability of DrBHV to serve as a vaccine vector for vampire bat rabies is not explicitly evaluated in this study. Rather, characteristics of this natural virus were described that may make it a candidate for further study on this topic.

>L246,248, 394: Check for typo in "*D. ecuadata*".

>L249: Rather than saying "other than oropharyngeal swabs," could you perhaps be specific and say "using blood samples"?

>L250-251: I think this point may also be argued for the other BHVs identified in this dataset. Host shifts in general are rare, so is it possible that increased sampling would recover some more evidence for this occurring in some of your other bat species?

>L285-289: Please explain why lower levels of detection in blood (vs. saliva) would be support for a persistent infection, and also why the virus during latency would be undetectable in blood, but present at low levels in saliva.

>L253: But this strain was also found in *A. lituratus*, so is not independently maintained in *D. rotundus*.

>L352, 384-385: Please provide information on the breakdown of saliva and blood sample sizes by bat species, perhaps in a Table or Supp Info. Could also provide actual sample sizes for the number of BHVs sequenced in *D. rotundus* and all other positive bat samples.

>Fig. 1: Could better highlight the focal species (*D. rotundus*) here, perhaps with bold font?

>Fig. 3: Is there a way to arrange or rotate nodes on the nucleotide and amino acid phylogenies such that the major taxa generally appear in a similar order (e.g., with primates on top and bats on bottom, or vice versa)?

Reviewer #3:

Remarks to the Author:

This is an interesting paper on the prevalence and spread of betaherpesviruses (BHV) in bats in the New (and Old) World. Specifically, the focus is on BHV in vampire bats (*Desmodus rotundus*) and species that share habitats with this main transmitter of bat lyssavirus in South America. The manuscript is straightforward and the results presented are compelling and interesting. The authors

find a remarkably high prevalence of BHV in *D. rotundus* (higher than 90% prevalence), less so in other species, where prevalences range from less than 1 to 30%. What is more, the BHV hosted by *D. rotundus* seem highly variable and a consequence of superinfections with different viruses.

The main weakness of the paper is that the authors claim, in fact that claim is in the title, that BHV are (a) good candidate(s) for delivering vaccine antigens to bats, specifically bat lyssavirus antigens and suggest that this theoretical possibility is now a reality - based on their findings. While the authors clearly show that BHV are variable (see above), there is absolutely no evidence presented that such vaccination would in fact be possible. Neither is the vaccine backbone identified nor is there any attempt made to generate such a vaccine.

In the following document, we present a point-by-point response to each reviewers' critique, with our responses in **bold** typeface.

REVIEWER COMMENTS

Reviewer #1 (Remarks to the Author):

In this paper, the authors report results from genetic surveys and phylogenetic analyses of a Betaherpesvirus, DrBHV, being investigated as a possible vector for a transmissible vaccine targeting rabies in vampire bats. In addition to presenting results evaluating the scope for DrBHV to serve as a useful vaccine vector for this specific application, the paper provides a template for a general approach to screening candidate vaccine vectors in other wildlife species. Primary results from the authors' analyses are that: 1) the prevalence of DrBHV is high within vampire bats, 2) multi-strain infections occur, and 3) host switching appears to be rare. These results are promising for the use of DrBHV as a vector for a transmissible vaccine as the authors argue. At the same time, as the authors recognize, this represents only an initial screen for suitability as a vector, not conclusive evidence that DrBHV will be optimal, or even suitable. Overall, this is an extremely clear paper that presents interesting results in a thoughtful and well-balanced way. My only substantive concern is the absence of information on rabies within these samples. In following paragraphs, I detail why I think this matters and also identify a few other issues that could be clarified or changed to improve the paper.

The focus of the work is identification of a vector suitable for developing a transmissible vaccine targeting rabies in vampire bats. Results are reported on the vector, but no information is provided on the incidence of rabies within the 128 vampire bats sampled. Given the authors' interest in rabies and background studying it in vampire bats, it seems likely this information was also collected from these samples and might be available. If so, the paper would be much stronger and the results easier to interpret if this information were provided. It matters because: 1) The presence of rabies in the same samples would demonstrate the distribution of DrBHV as measured here is relevant for rabies control, 2) The pattern of rabies infection relative to DrBHV infection would shed light on whether both viruses follow similar infection pathways, and 3) The prevalence of both viruses could be compared in a meaningful way, allowing for more concrete predictions about the likely performance of a transmissible vaccine targeting rabies using DrBHV.

We agree that information regarding the prevalence and distribution of rabies virus in vampire bats would lend strength to the argument that a vaccine is necessary. We further note that such information is relevant to the suitability of transmissible vaccines in that the primary criteria for eradication is that the vaccine should be more transmissible than the target virus. As such, we have tested 123 of the 128 saliva samples used in the manuscript by PCR for rabies virus. We have also collated relevant rabies serology data for the years and departments covered in this study in order to demonstrate that vampire bat transmitted rabies virus (VBRV) is widely distributed around Peru, and shares high correlation to concurrent outbreaks of rabies in cattle. These new results show that rabies is widespread in Peru, but has relatively low transmissibility, infecting <1% of bats we tested by RT-PCR, and a seroprevalence of ca. 10%. Notably, mathematical models and experimental infections show that these naturally seropositive individuals are likely to have some degree of protection against future rabies exposures, further contributing to the feasibility of using vaccines for rabies control. In the revised manuscript, these results are discussed on lines 165-180(results), 320-334 (discussion) and Figure 2.

Lines 76-78: "Furthermore, if vaccine-induced mortality were to occur in the reservoir host, this would not only pose animal welfare concerns, but also would have the undesirable effect of

reducing vaccine-induced immunity at the population level.” This may not be as relevant as it sounds because death also removes individuals from the susceptible pool which curtails the spread of the infectious disease, at least if transmission is density dependent.

This line has been removed due to lack of relevance as per suggestion.

I agree that the existence of multiple strains within a bat suggests reinfection. Unfortunately, for recombinant vector transmissible vaccine what really matters is whether reinfection with an identical strain is possible. This matters for two reasons. First, in most cases, the vector would be a strain already present within the target population and so the vaccine will compete with its progenitor unless it can re infect/superinfect. It might be possible to get around this issue by using a rare or infrequent strain as a backbone, but this comes at the risk of choosing a strain that is rare because it is not terribly transmissible/competitive in the first place. Second, no matter how good the engineering, the vaccine will almost certainly evolve to lose or downregulate its immunogenic cargo. Once it does, intact vaccine must compete with these mutants unless reinfection/superinfection by identical strains is possible. I think these more subtle issues need to be better developed to place the authors’ very cool (but not comprehensive) results on reinfection/superinfection in their full context.

This is an excellent point that we agree deserves to be further developed. In the revised manuscript, we discuss – based on previous work on the population genetics of both bat hosts and rabies virus - the strong possibility that multiple strains of DrBHV circulate around Peru with differing prevalence by region. As the reviewer suggests, this opens the possibility that a strain that circulates at high levels in one region (and therefore has the capacity to spread well in an area if introduced) could be used for the vaccine vector in another area. Lines 358-370 have been added to reflect such.

The second point about competition of vaccine strains with previous vaccine strains that have lost their insertion is absolutely valid and we agree needs to be evaluated prior to vaccine releases. We have suggested how this could be addressed via experimental infections in Lines 370-373.

Reviewer #2 (Remarks to the Author):

Summary:

The objective of this study was to identify a non-pathogenic virus as a candidate for use in designing a transmissible vaccine against vampire bat rabies virus. Specifically, the study evaluates the suitability of a novel betaherpesvirus (BHV) in vampire bats (*D. rotundus*), by assessing BHV prevalence, host specificity, and co-evolution across a number of bat species and other mammalian hosts (i.e., rodents and primates). The authors conclude that a BHV strain in vampire bats (DrBHV) is a good candidate for a viral vaccine, due to their findings that it is found at high prevalence and the evolutionary history of BHVs suggest host-specificity. While the idea of using virus as a vector for wildlife vaccination is certainly cutting-edge and exciting, after reading the article, I found that this spin was slightly misleading with regards to the actual data, analyses, and results. Though the vaccine angle does provide justification for this work. Further detailed comments and suggested edits may be found below.

Major Comments:

>While, I do think the analyses are valid and provide some interesting information, the data are not explicitly testing the use of DrBHV as a potential vaccine vector (as the title and initial introduction imply), but rather describe the prevalence and distribution of a virus in bats where little previous information is currently available. I can see how the vaccine angle is justification for the study and should be emphasized somewhere in the paper, but think the authors may be overstating the potential for this (at this time) as a natural vector for a bat rabies vaccine.

As per suggestion, the transmissible vaccine angle of the manuscript has been toned down. This includes changes in the title and abstract to remove any unintended implication that a transmissible vaccine had been designed/tested. We further re-organized the introduction to focus more on the importance and management challenges of vampire bat rabies. The vaccination angle therefore raised later in the introduction, rather than being the primary focus of the manuscript. We have also been as clear as possible throughout that we are testing the ecological, biological and epidemiological prerequisites for DrBHV as a vaccine vector, but that we are not currently evaluating an engineered form of this virus (see Lines 130-131 and Lines 271-275). We believe this re-structuring of the manuscript, together with the new data presented on rabies creates a reasonable balance of the reviewer's concern and the aims of our manuscript, which were based on testing prerequisites for a transmissible vaccine vector.

>Along these same lines, the authors' claim the prevalence and host-specificity of BHV is evidence that the virus would be a suitable vector. But, the data seem to suggest a long evolutionary history of the virus with the host and it is not clear what the transmission rate would be (i.e. how long did it take for the natural strains of BHV to become established). How would an introduced recombinant virus achieve the same level of distribution and prevalence? There is an assumption here that emergence and persistence will be likely, but the time frame of the natural emergence process is unknown. And, their data also suggest multiple co-circulating BHV strains (or super-infections), so it is possible there are several viral strains with variable distributions and prevalence.

Whilst it is highly likely that the timescale of natural BHV evolution was long, we do not believe that an introduced recombinant vaccine would necessarily face the same time constraints. Artificial introduction of a vaccine strain can take place in multiple locations, with hundreds of individuals inoculated. This scale of virus introduction should accelerate the rate by which a sustained viral prevalence is reached within the bat population. Furthermore, we see no reason that viruses which co-speciated with their hosts on evolutionary timescales cannot also be highly transmissible on epidemiological timescales, which is what matters most for vaccine spread. The matter of multiple virus strains and the timescale of introduction has been expanded on (Lines 378-381).

>L162-163: I'm not sure if there are enough samples per species here to really evaluate host specificity and degree of host shifts at the species level. The statement that there are host species-specific clades is not entirely supported with the fact that *A. lituratus* clusters with *D. rotundus*, though this is mentioned later in the paragraph. Could more convincingly support family-level clustering and higher level taxonomic co-evolution. But, again, I'm not convinced sampling is comprehensive enough to detect the variability in BHVs that may be present across bat host species. Do some sequences in the nucleotide-based phylogeny represent multiple samples? If so, this could be more clearly presented somewhere.

Some sequences in the phylogeny did indeed represent multiple samples which clustered together, of the same species. These were originally excluded from the figure to save space, but have now been added in to address the reviewer's point. Specifically, 1 extra *Myotis*, 2 extra *G.*

soricina, and 11 extra *D. rotundus* were added to the new figure. The lack/low level of sampling in non-*D. rotundus* species is a limitation which is now acknowledged in the manuscript (Lines 290-293). Specifically, as suggested, we focus on family level clustering and emphasize that a greater sampling effort is required to definitively describe the host specificity of BHVs in bats.

>L264-266: As commented on above, the time frame for establishing high prevalence of a transmissible vaccine may be quite long. While it is not feasible with these data, perhaps it would be possible to do a coalescent analysis (with genome-level BHV sequence data from vampire bats) to estimate the time to the most recent ancestor, in order to predict the time it would take for a vaccine developed from this virus to reach similarly high prevalence. Though, given the release parameters (number and distribution released), this may be paired with a modelling exercise or focused on a specific population as a case study. Again, these are just suggestions for future work, as I realize this is not the aim of the current paper.

We agree with that establishing high prevalence is crucial. Indeed, extensive modelling of DrBHV transmission parameters both within a bat colony and over a larger scale across Peru are planned. However, we are reluctant to draw direct parallels between historical emergence and vaccine releases for two main reasons. First, this virus putatively emerged at the time that *D. rotundus* speciated. All relatives from the genus *Desmodus* are now extinct and exact dates of speciation are unknown, but its closest extant relative occurred approximately 23 million years ago. It therefore is certain that the environmental conditions under which this virus originally emerged are incomparable to how an introduced virus would spread in the present day. This is especially relevant given that *Desmodus* populations have expanded both in number and geographic range along with the proliferation of livestock in Latin America since European colonization. Second, and as mentioned in the manuscript (Lines 368-369 and 378-381), a vaccine release could take advantage of multiple vaccine strains and multiple release locations/dates. The phylogenetic approach suggested would be particularly insightful to guide such releases, but we think this needs to be done on relatively small spatial and temporal scales to better approximate vaccine spread. We are currently generating datasets to do exactly this.

Minor Comments and Suggested Revisions:

>Paragraph starting L143: In the text, where are the results shown in Fig. 1 comparing infection prevalence across species?

Statistics for the species level glm can be found in the supplementary materials (table 1).

>L152: How many species are represented in the BHV prevalence value from blood samples? Is this just *D. rotundus* or other bat species as well?

Only *D. rotundus* blood samples were tested for DrBHV. This has been clarified in the text at Line 159.

>L153-154: I think this is an interesting discussion point. Does this mean that ~55% of positive samples (from saliva samples) could be due to environmental contamination? With a possible systemic infection prevalence as low as 45%, would this be sufficient for a successful vaccination campaign? Typically larger vaccination coverage is needed to reduce transmission, and for herd immunity.

Whilst it is possible that environmental contamination is the culprit, this is unlikely. It is expected that saliva PCRs would have a greater prevalence of BHV than blood, as viremia may not occur in all infected individuals, or may take some time after infection when shedding in saliva is already present. This is frequently observed in HCMV patients. We have added further explanation for our reasoning at Lines 160-163.

>L186-7, 191: Is “P=0” correct, or should it be written as “P<0.0001”?

This has been amended (Lines 216-218, 223-224).

>L204: By “different,” do you mean “multiple unique strains” in both samples?

This statement was intended to mean multiple unique strains; this has been amended in the text (Line 243).

>L212-213: Should this be 2-3 major strains at each location (for sample 10148)? Panel B appears to indicate 3 strains, or are you considering the top cluster as a single major strain?

We are considering the top cluster as a single major strain, due to a lower than 10% sequence difference between the two, albeit very short, sequences.

>L232-238: The ability of DrBHV to serve as a vaccine vector for vampire bat rabies is not explicitly evaluated in this study. Rather, characteristics of this natural virus were described that may make it a candidate for further study on this topic.

We apologize if our original wording was misleading. The reviewer’s wording reflects our own sentiment and intended meaning. Throughout the revised text we have checked to ensure that the wording on this point is clear.

>L246,248, 394: Check for typo in “D. ecuadata”.

The spelling has been corrected to “ecaadata” in all cases.

>L249: Rather than saying “other than oropharyngeal swabs,” could you perhaps be specific and say “using blood samples”?

This change has been made, Line 297.

>L250-251: I think this point may also be argued for the other BHVs identified in this dataset. Host shifts in general are rare, so is it possible that increased sampling would recover some more evidence for this occurring in some of your other bat species?

It is quite possible that greater sampling of bats would uncover more evidence of host shifts. Whilst we can only speculate on this front, the caveat that undetected host shifts or cross-species transmission events are possible has been added to the manuscript, Lines 290-293.

>L285-289: Please explain why lower levels of detection in blood (vs. saliva) would be support for a persistent infection, and also why the virus during latency would be undetectable in blood, but present at low levels in saliva.

Thank-you for raising this issue, it appears that two separate points have been confused in the manuscript; a lower level of detection in blood vs saliva supports a multi-stage infection as viremia may take place months after initial infection and active shedding in saliva, and latently infected monocytes in the peripheral blood may be detected via PCR years after initial infection. Low levels of detection in saliva may indicate a latent infection. We have not yet tested enough paired blood and saliva samples to judge whether presence or absence of viral DNA in blood correlates with level of detection in the saliva PCR. qPCR tests are currently planned for further determination of viral load in saliva and blood over time. This has been clarified and added to the text (Lines 162-163 and 343-346).

>L253: But this strain was also found in *A. lituratus*, so is not independently maintained in *D. rotundus*.

DrBHV can be found outside of the range of *A. lituratus*, so this species is not necessary for the long-term maintenance of the virus. Whilst the *A. lituratus* sample suggests that cross-species transmission within Phyllostomidae can take place or a possible instance of contamination due to ecological interactions between these species, we believe it is accurate to say that *D. rotundus* can independently maintain DrBHV.

>L352, 384-385: Please provide information on the breakdown of saliva and blood sample sizes by bat species, perhaps in a Table or Supp Info. Could also provide actual sample sizes for the number of BHVs sequenced in *D. rotundus* and all other positive bat samples.

A table containing this information has been added to the supplementary materials (Supp. Table 2).

>Fig. 1: Could better highlight the focal species (*D. rotundus*) here, perhaps with bold font?

***D. rotundus* has been appropriately highlighted.**

>Fig. 3: Is there a way to arrange or rotate nodes on the nucleotide and amino acid phylogenies such that the major taxa generally appear in a similar order (e.g., with primates on top and bats on bottom, or vice versa)?

Whilst this is technically possible, the nodes have been automatically rotated so that the links between them are as clear as possible. We believe that a rearrangement would impede rather than clarify understanding of the figure.

Reviewer #3 (Remarks to the Author):

This is an interesting paper on the prevalence and spread of betaherpesviruses (BHV) in bats in the New (and Old) World. Specifically, the focus is on BHV in vampire bats (*Desmodus rotundus*) and species that share habitats with this main transmitter of bat lyssavirus in South America. The manuscript is straightforward and the results presented are compelling and interesting. The authors find a remarkably high prevalence of BHV in *D. rotundus* (higher than 90% prevalence), less so in other species, where prevalences range from less than 1 to 30%. What is more, the BHV hosted by *D. rotundus* seem highly variable and a consequence of superinfections with different viruses.

The main weakness of the paper is that the authors claim, in fact that claim is in the title, that BHV are (a) good candidate(s) for delivering vaccine antigens to bats, specifically bat lyssavirus antigens

and suggest that this theoretical possibility is now a reality - based on their findings. While the authors clearly show that BHV are variable (see above), there is absolutely no evidence presented that such vaccination would in fact be possible. Neither is the vaccine backbone identified nor is there any attempt made to generate such a vaccine.

We did not intend to imply that a vaccine had been created or tested in practice, only that DrBHV meets some key prerequisites and should be targeted for further study. We have reframed the manuscript in order to remove any potentially misleading claims, including changing the title of the manuscript, and rearranging the introduction to de-emphasize transmissible vaccines.

We have also included in the results a further evaluation of the viral genome organisation and how it might serve as a future vaccine backbone to enhance our findings (Lines 260-265).

Reviewers' Comments:

Reviewer #1:

Remarks to the Author:

The authors have heavily revised their manuscript in response to the comments of the reviewers and have addressed the suggestions I made in my original review. I really appreciate the authors' willingness to include the Rabies data I requested. I believe this paper makes a very important and solid contribution to emerging efforts to develop transmissible vaccines. I have only a few remaining suggestions for improvement which I detail below.

I did not see the extent to which infection by VBRV and DrBHV overlap reported. Were VBRV seropositive bats ever infected with DrBHV? Perhaps this is asking too much, but it would maximize the future utility of this work if the data on DrBHV and VBRV infection were made publicly available either as a table in the paper or as a supplemental file (e.g., .csv) organized something like: Bat ID, Location of capture, Date of capture, DrBHV PCR status, VBRV PCR status, and VBRV Serostatus.

Why were only two PCR positive samples subjected to metagenomic sequencing and explored for evidence of super-infection? How were these samples chosen? Specifically, were they chosen at random or were they selected because they had particularly high viral titers or some other feature that suggests they may not be representative of the remaining samples? I understand there are certainly financial constraints on what can be accomplished here, but convincingly demonstrating that superinfection is common (rather than just possible) would require that additional samples be subjected to metagenomic sequencing. Recognizing it is unrealistic to ask the authors to do additional sequencing at this point, explaining how/why these particular samples were chosen and if they are likely to be representative is adequate.

It is not entirely clear to me how within host evolution can be ruled out and super-infection supported based on the available data. Can the authors expand a bit on how we know sequence divergence is too great to be explained by within host evolution? Specifically, given DrBHV presumably generates lifelong chronic infections of long-lived animals, would we not perhaps expect to see considerable within host evolution? How much viral diversification would we expect to see over, say, 5-10 years of infection? To be clear, I trust the authors interpretation, I would just like to see a bit more justification/explanation for the conclusion in the text.

In summary, this is a very nice piece of work which should be accepted for publication. Addressing any of the issues raised above through further revision would improve the paper but is not essential.

Reviewer #2:

Remarks to the Author:

I am happy with the edits that the authors made to this paper to address my comments, as well as those of the other reviewers. In particular, the RABV data on infection prevalence and distribution is a great addition, and help make a stronger link to the potential use of this newly identified herpesvirus as a transmissible vaccine for rabies. I also like the reframing of the introduction, and think it provides a nice framework for the initial process of identifying a virus candidate for a wildlife vaccine. The study and its implications are quite exciting and promising, and I think it will generate a lot of interest. I have included a couple minor comments below.

Minor Comments:

>Abstract: Include a sentence here on the newly added RABV rt-PCR infection and serology results. In the paragraph starting on L130, you could also indicate that RABV prevalence and distribution were examined to determine the potential use of a BHV vector for targeting rabies.

>Figure legends are missing in this submission. This was only a problem for the new Figure 2, as I could refer to the previous version for the others. In any case, I get the general idea being communicated in the figure. However, it is hard to see how positive versus negative samples are spatially distributed. I assume closed circles mean positive and open negative? You could also outline region areas on the map that encompass the samples included in each of the plots.

Response to reviewers' comments:

Our response to reviewers can be found in **bold**.

Reviewer #1 (Remarks to the Author):

The authors have heavily revised their manuscript in response to the comments of the reviewers and have addressed the suggestions I made in my original review. I really appreciate the authors' willingness to include the Rabies data I requested. I believe this paper makes a very important and solid contribution to emerging efforts to develop transmissible vaccines. I have only a few remaining suggestions for improvement which I detail below.

I did not see the extent to which infection by VBRV and DrBHV overlap reported. Were VBRV seropositive bats ever infected with DrBHV? Perhaps this is asking too much, but it would maximize the future utility of this work if the data on DrBHV and VBRV infection were made publicly available either as a table in the paper or as a supplemental file (e.g., .csv) organized something like: Bat ID, Location of capture, Date of capture, DrBHV PCR status, VBRV PCR status, and VBRV Serostatus.

In order to clarify the comparison between BHV and VBRV infection testing, we have created a table in the format suggested, which is now included as part of the supplementary materials (supplementary table 4). In response to the reviewer's specific query on whether seropositive bats were infected with DrBHV, we report 12 instances of this (12% of rabies-seropositive bats that were tested for DrBHV). This is unsurprising given that nearly all bats in the populations we studied were DrBHV-positive.

Why were only two PCR positive samples subjected to metagenomic sequencing and explored for evidence of super-infection? How were these samples chosen? Specifically, were they chosen at random or were they selected because they had particularly high viral titers or some other feature that suggests they may not be representative of the remaining samples? I understand there are certainly financial constraints on what can be accomplished here, but convincingly demonstrating that superinfection is common (rather than just possible) would require that additional samples be subjected to metagenomic sequencing. Recognizing it is unrealistic to ask the authors to do additional sequencing at this point, explaining how/why these particular samples were chosen and if they are likely to be representative is adequate.

As the reviewer guessed, we limited our metagenomic sequencing to only two samples due to budgetary constraints and the sequencing depth needed to cover the expected large size of the DrBHV genome by shotgun sequencing. One sample was chosen due to its strong PCR band, and therefore likelihood that sequencing would be successful in this sample. The second sample was not as strongly positive and was more representative of the rest of the samples tested by PCR. This has been elaborated upon in the methods (L537-539).

It is not entirely clear to me how within host evolution can be ruled out and super-infection supported based on the available data. Can the authors expand a bit on how we know sequence divergence is too great to be explained by within host evolution? Specifically, given DrBHV presumably generates lifelong chronic infections of long-lived animals, would we not perhaps expect to see considerable within host evolution? How much viral diversification would we expect to see over, say, 5-10 years of infection? To be clear, I trust the authors interpretation, I would just like to

see a bit more justification/explanation for the conclusion in the text.

The distinction between super-infection and intra-host evolution has been expanded upon in the discussion (L350-355), with supplementary figure 2 illustrating what could be interpreted as two completely different strains in comparison to potential evolution of each strain. We have also discussed similar work carried out in HCMV, which theoretically has a much larger time frame in which to undergo evolution within a host, given the lifespan of humans.

In summary, this is a very nice piece of work which should be accepted for publication. Addressing any of the issues raised above through further revision would improve the paper but is not essential.

We thank the reviewer for the constructive comments.

Reviewer #2 (Remarks to the Author):

I am happy with the edits that the authors made to this paper to address my comments, as well as those of the other reviewers. In particular, the RABV data on infection prevalence and distribution is a great addition, and help make a stronger link to the potential use of this newly identified herpesvirus as a transmissible vaccine for rabies. I also like the reframing of the introduction, and think it provides a nice framework for the initial process of identifying a virus candidate for a wildlife vaccine. The study and its implications are quite exciting and promising, and I think it will generate a lot of interest. I have included a couple minor comments below.

Minor Comments:

>Abstract: Include a sentence here on the newly added RABV rt-PCR infection and serology results. In the paragraph starting on L130, you could also indicate that RABV prevalence and distribution were examined to determine the potential use of a BHV vector for targeting rabies.

These sentences have been added in the relevant sections.

>Figure legends are missing in this submission. This was only a problem for the new Figure 2, as I could refer to the previous version for the others. In any case, I get the general idea being communicated in the figure. However, it is hard to see how positive versus negative samples are spatially distributed. I assume closed circles mean positive and open negative? You could also outline region areas on the map that encompass the samples included in each of the plots.

Apologies for the lack of figure legends, they were misplaced in the file upload procedure. The map figure shows positive cases of VBRV in livestock over the course of three years. The region areas on the map corresponding to 2a-f have been shaded so that they are easier to identify.